# VideoEval-Pro: Robust and Realistic Long Video Understanding Evaluation

## Abstract

Large multimodal models (LMMs) have recently emerged as a powerful tool for long video understanding (LVU), prompting the development of standardized LVU benchmarks to evaluate their performance. However, our investigation reveals a rather sober lesson for existing LVU benchmarks. First, most existing benchmarks rely heavily on multiple-choice questions (MCQs), whose evaluation results are inflated due to the possibility of guessing the correct answer; Second, a significant portion of questions in these benchmarks have strong priors to allow models to answer directly without even reading the input video. For example, Gemini-1.5-Pro can achieve over 50% accuracy given a random frame from a long video on Video-MME. We also observe that increasing the number of frames does not necessarily lead to improvement on existing benchmarks, which is counterintuitive. As a result, the validity and robustness of current LVU benchmarks are undermined, impeding a faithful assessment of LMMs' long-video understanding capability. To tackle this problem, we propose VideoEval-Pro, a realistic LVU benchmark containing questions with open-ended short-answer, which truly require understanding the entire video. VideoEval-Pro assesses both segment-level and full-video understanding through perception and reasoning tasks. By evaluating 27 proprietary and open-source video LMMs, we conclude the following findings: (1) video LMMs show drastic performance (>25%) drops on open-ended questions compared with MCQs; (2) surprisingly, higher MCQ scores do not lead to higher open-ended scores on VideoEval-Pro; (3) compared to other MCQ benchmarks, VideoEval-Pro benefits more from increasing the number of input frames. Our results show that VideoEval-Pro offers a more realistic and reliable measure of long video understanding, providing a clearer view of progress in this domain. Our benchmark and evaluation code will be fully released.

## 1 Introduction

Long video understanding (LVU) refers to the task of using AI systems to process, interpret, and reason over long-duration video content. Key applications of long video understanding include event and anomaly detection in video surveillance (Lv and Sun, 2024), temporal reasoning and behaviour prediction in autonomous driving (Ettinger et al., 2021), as well as content summarization in instructional/lecture videos (Zhou et al., 2018). Designing AI systems capable of understanding and reasoning over long videos is therefore a fundamental challenge in artificial intelligence.

Recently, large multimodal models (LMMs) have emerged as a promising solution for understanding long videos. Ongoing research enhances the capability of LMMs to process longer videos by extending their context length (Zhang et al., 2024a; Chen et al., 2024a), dropping or merging video tokens (Li et al., 2024a; Wang et al., 2025a), and leveraging efficient, linear-complexity models (Ren et al., 2025; Jiang et al., 2025; Islam et al., 2025). Besides model architecture improvements, recent studies also investigate curating better training data (Zhang et al., 2024b; Ren et al., 2024) and applying reinforcement learning approaches (Li et al., 2025; Feng et al., 2025) for LMMs tailored to LVU tasks. As a result, LMMs have rapidly advanced to achieve stronger LVU capabilities: the preliminary attempt of Video-LLaVA (Lin et al., 2023) can only process short videos with eight frames. Today, LMMs such as Vamba (Ren et al., 2025), Video-XL-Pro (Liu et al., 2025), and InternVideo2.5 (Wang et al., 2025a) can encode thousands of frames and reason over hour-long videos.

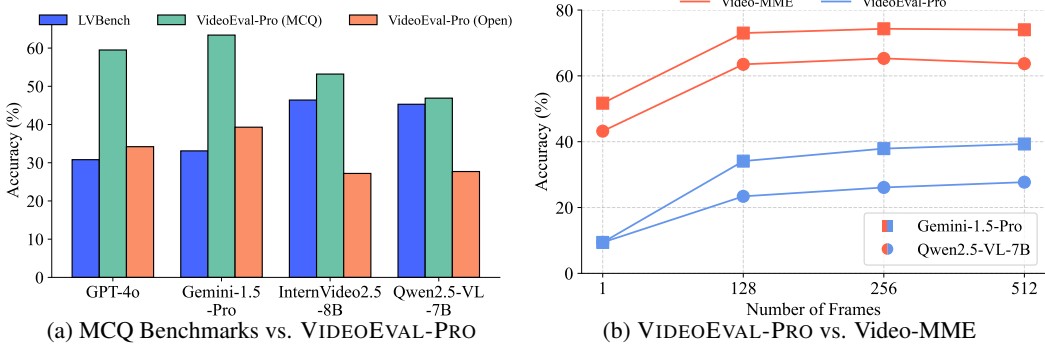

Figure 1: Comparison between VIDEOEVAL-PRO and MCQ benchmarks. **Left:** MCQ benchmarks yield inflated scores on identical questions (MCQ vs. Open) and can misrepresent model performance (LVBench (Wang et al., 2024a)). **Right**: VIDEOEVAL-PRO cannot be effectively solved with a single input frame, and performance scales consistently with more frames. Video-MME (Fu et al., 2024) exhibits contradictory trends.

To rigorously evaluate ongoing advances in video LMMs, researchers have introduced dedicated long video understanding benchmarks (Fu et al., 2024; Zhou et al., 2024; Wang et al., 2024a; Wu et al., 2024; Chandrasegaran et al., 2024), which provide standardized scores to quantitatively measure and compare different models' ability to reason over long videos. Nevertheless, upon examining the current LVU benchmarks more closely, our findings are sobering. First, most current LVU benchmarks rely almost exclusively on multiple-choice questions (MCQs), a format that can inadvertently provide hints to the model, enabling it to answer correctly through guesswork. As shown in Figure 1a, for the same set of questions, we observe over 20% accuracy drop when switching from MCQ to open-ended question answering. This significant gap suggests that MCQ-based accuracies may be substantially inflated and do not reliably reflect the model's true understanding of the video content. Second, many questions in existing LVU benchmarks exhibit strong priors that allow models to answer correctly without comprehending the input video. As illustrated in Figure 1b, both proprietary (Gemini-1.5-Pro (Team et al., 2024)) and open-source (Qwen2.5-VL-7B (Bai et al., 2025)) models achieve around 50% accuracy on Video-MME (Fu et al., 2024) with only one input frame. These issues lead to performance plateauing or even declining as input frames increase—an outcome that contradicts the expectation that more frames should offer richer context and improve long video understanding. Our investigations of existing LVU benchmarks prompt us to think about two central questions:

(1) **Do existing long video benchmarks faithfully reflect models' real capacity to understand long video content?** (2) **Do the gains reported by newer models genuinely translate into stronger long video comprehension capability, or are they illusional?**

To probe these questions, we present VIDEOEVAL-PRO, a more robust and realistic long video understanding benchmark containing open-ended, short-answer QA problems. To construct VIDEOEVAL-PRO, we source the questions from four existing long video understanding MCQ benchmarks: Video-MME (Fu et al., 2024), MLVU (Zhou et al., 2024), LVBench (Wang et al., 2024a) and LongVideoBench (Wu et al., 2024), and reformat these questions into free-form questions. We apply a series of filtering methods based on video duration, question and answer type, answerability and QA difficulty to ensure the quality. Our final benchmark contains a total of 1,289 short-answer questions based on 465 videos, with an average duration of 38 minutes. By evaluating a total of 27 proprietary and open-source models, our main findings can be summarized as below:

1. VIDEOEVAL-PRO and its open-ended QA format introduce substantial challenges for video LMMs, as evidenced by performance drops exceeding 25% compared to the MCQ format.
2. VIDEOEVAL-PRO's results show that LMMs perform better at questions about local video segments and struggle more with those requiring holistic video understanding. They also perform better on perception tasks than on reasoning tasks.
3. Unlike existing MCQ benchmarks, VIDEOEVAL-PRO reveals that proprietary models still hold a significant lead, indicating the brittleness of open-source models on challenging LVU tasks.
4. Current LMMs achieve only ∼10% accuracy on VIDEOEVAL-PRO with a single input frame, but performance steadily improves with more frames. These results highlight our benchmark's requirement for rich temporal information and show that VIDEOEVAL-PRO is a more suitable benchmark for long video understanding.

## 2 RELATED WORK

### 2.1 LARGE MULTIMODAL MODELS FOR LONG VIDEO UNDERSTANDING

The recent growing demand for long video understanding has driven the rapid evolution of large multimodal models (LMMs). Prior methods such as Video-ChatGPT (Maaz et al., 2023) and Video-LLaVA (Lin et al., 2023) focus on understanding short videos based on limited input frames (e.g. 8 frames). Since then, the development of video LMMs generally follows two directions. First, prior work investigates various model architecture improvements to enable LMMs to process more input frames. For example, LongVU (Shen et al., 2024), Video-XL (Shu et al., 2024), InternVideo2.5 (Wang et al., 2025a) and VideoChat-Flash (Li et al., 2024a) investigates token dropping to reduce sequence length. STORM (Jiang et al., 2025), Vamba (Ren et al., 2025) and BIMBA (Islam et al., 2025) utilize hybrid architectures (Gu and Dao, 2023) to enable more efficient video processing. Second, recent work also investigates methods to improve the training of video LMMs. LLaVA-Video (Zhang et al., 2024b), Vript (Yang et al., 2024) and VISTA (Ren et al., 2024) collect higher-quality training data to enhance video LMM training. LLaVA-Hound-DPO (Zhang et al., 2024c), Video-R1 (Feng et al., 2025) and VideoChat-R1 (Li et al., 2025) investigate preference optimization and reinforcement learning techniques to enhance the reasoning capabilities of video LMMs.

### 2.2 LONG VIDEO UNDERSTANDING BENCHMARKS

The rapid development of video LMMs has spurred the creation of video understanding benchmarks, aiming at evaluating video LMM's perception and reasoning capabilities based on video inputs. Earlier video QA benchmarks such as MSRVTT-QA/MSVD-QA (Xu et al., 2017) and ActivityNet-QA (Yu et al., 2019) focus on short video clips with simple questions. MVBench (Li et al., 2024b) constructs a unified benchmark by regenerating QA pairs from existing datasets, while TempCompass (Liu et al., 2024) and VideoVista (Li et al., 2024c) focus on assessing temporal reasoning. More recently, LVU benchmarks such as Video-MME (Fu et al., 2024), MLVU (Zhou et al., 2024), LVBench (Wang et al., 2024a), and LongVideoBench (Wu et al., 2024) have emerged to evaluate the performance of LMMs on extremely long videos using MCQ questions. However, the potential influence of MCQ options, such as providing answer hints or inducing bias in LVU evaluation, has not been systematically studied. To bridge this gap, we aim to introduce a new benchmark featuring concise free-form answers, aiming to better reflect models' true LVU capability without relying on pre-defined choices.

## 3 VIDEOEVAL-PRO

### 3.1 DATA CURATION PIPELINE

VIDEOEVAL-PRO's data curation pipeline comprises two main steps: Data Collection and Data Filtering. In this section, we describe each step in detail.

**Data Collection** To construct VIDEOEVAL-PRO, we first collect source QA pairs from four publicly available long video understanding benchmarks: Video-MME (Fu et al., 2024), MLVU (Zhou et al., 2024), LVBench (Wang et al., 2024a) and LongVideoBench (Wu et al., 2024). These benchmarks span diverse video content and question types, providing a rich source for long video understanding tasks. Our initial seed question set contains a total of 5,562 questions, which are all in MCQ format with 4-6 options. To create an open-ended evaluation benchmark, we transform each multiple-choice question into a free-form question: the correct MCQ option becomes the reference answer, while the distractors are discarded. During evaluation, the models receive only the question itself, forcing them to generate an answer based on the input video rather than exploiting hints from different options.

**Video Duration and Answer Type Filtering** Once the initial pool of questions is collected, we apply a multi-stage filtering process to ensure that the resulting dataset emphasizes long-term video comprehension and presents a meaningful challenge for current models. We first filter out all samples associated with videos shorter than 10 minutes, as shorter clips often contain less complicated contents that may lower the difficulty of video perception and reasoning tasks. Next, we remove questions for which the average word count of answer options in the original MCQ format exceeds five words. For example, questions such as "What is this video about?" often yield overly detailed responses, which complicates answer evaluation. This word-count constraint reduces uncertainty

from overly verbose options and ensures that the converted open-ended questions have concise yet meaningful answers, making it easier for LLM judges (*c.f.* Section 3.3) to evaluate model responses, thereby enhancing the stability and precision of our benchmark and improving its overall validity.

**Answerability Filtering**  In the next stage, we assess whether each multiple-choice question can be reasonably reformulated into a free-form question without losing clarity or answerability. From the question pools we collected, we notice three types of questions with low answerability: (1) Option-evaluating or comparing questions, which require the model to compare different options and pick the most reasonable option; (2) Timestamp-dependent questions, which requires the model to answer questions for a given numerical timestamp; (3) Subtitle-dependent questions, which queries information that only appeared in the subtitles. We prompt Gemini-2.0-Flash with the question (excluding the answer choices) and ask it to determine whether the question can be answered solely based on the video content. This step helps identify and discard questions that rely heavily on inspecting MCQ options, which are unsuitable for open-ended evaluation.

**Difficulty Filtering**  Finally, we filter out questions that are too easy to answer. To identify such cases, we randomly sample a single frame from each input video and prompt Gemini-2.0-Flash to generate an answer to the corresponding MCQ and open-ended question using only that frame. We then use Gemini-2.0-Flash to judge the open-ended answers. Questions for which Gemini-2.0-Flash produces a correct response on both MCQ and open-ended formats are excluded from the benchmark. This filtering step ensures that the remaining questions require broader temporal understanding and cannot be resolved using minimal visual context.

## 3.2 DATASET STATISTICS

Our rigorous data collection and filtering pipeline ensures that the final benchmark questions demand deeper temporal comprehension and reasoning beyond surface-level cues. Our final dataset comprises 1,289 question-answer pairs in free-form QA, each grounded in a long video with a duration greater than 10 minutes. As shown in Table 1, VIDEOEVAL-PRO includes a total of 465 videos, with an average length of 38.25 minutes. Among them, 204 videos are between 10 and 30 minutes and 261 videos exceed 30 minutes. For the 1,289 questions used in our benchmark, 371 are associated with videos in the 10–30 minute range, while 918 are based on videos longer than 30 minutes. The average length of an answer is 2.1 words. These design choices ensure the evaluation focuses on the model's ability to retrieve concise and accurate information from long video content.

Table 1: Video and QA statistics for VIDEOEVAL-PRO.

| VIDEOEVAL-PRO | Total | 10–30 min | >30 min | Note |
|---|---|---|---|---|
| Videos | 465 | 204 | 261 | Average video duration: 38.25 minutes |
| QA Pairs | 1,289 | 371 | 918 | Average answer length: 2.1 words |

**QA Source Distribution**  As mentioned in Section 3.1, the questions in VIDEOEVAL-PRO are drawn from four existing benchmarks: Video-MME (Fu et al., 2024), MLVU (Zhou et al., 2024), LVBench (Wang et al., 2024a), and LongVideoBench (Wu et al., 2024). Given the variability in video duration and question quality across these sources, their contributions to the final dataset differ. As illustrated in Figure 2a, LVBench accounts for the largest portion, contributing 714 questions (55%) due to its long and information-rich video sources. MLVU contributes 267 questions (21%), with many QAs excluded because of the relatively short video lengths. Video-MME adds 272 questions (21%), although a significant number were filtered out due to limited answerability. LongVideoBench, which is smaller in scale and subject to strict selection criteria, contributes 36 questions (3%). This diverse composition ensures that VIDEOEVAL-PRO spans a wide range of content domains and video types, ensuring a comprehensive model evaluation.

**Task Definition and Distribution**  Given the questions we collected, we propose a unified and generalizable task taxonomy to categorize our benchmark questions into four main types and 15 subtypes. These task types capture both perception and reasoning demands for both local video segments and holistic long video understanding tasks. The four main task types are:

- **Local Perception (LP)**: LP focuses on identifying and retrieving visual elements or actions from a short video clip in a long video.

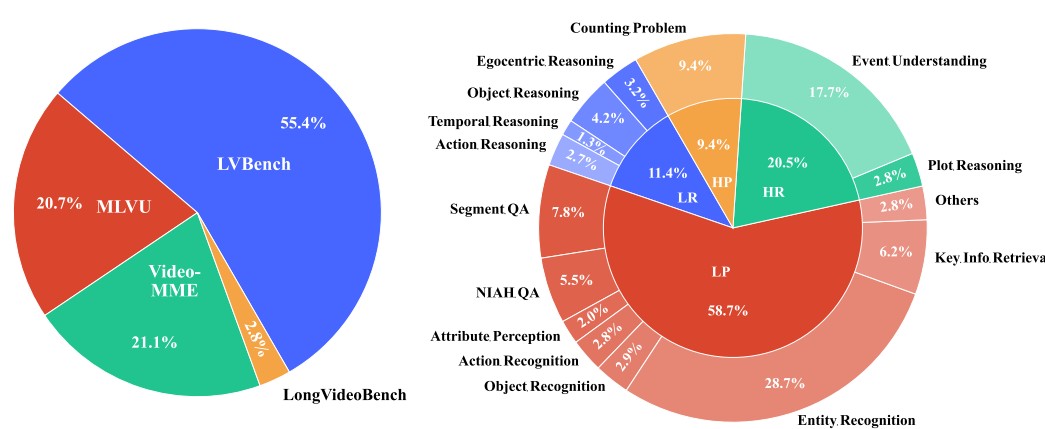

(a) Video and QA source distribution of VIDEOEVAL-PRO

(b) Task type distribution of VIDEOEVAL-PRO

Figure 2: Summary of VIDEOEVAL-PRO data composition and task type distribution.

- **Local Reasoning (LR)**: LR focuses on reasoning within short temporal windows, such as inferring causality, temporal order, or changes that happen over a local sequence of events.
- **Holistic Perception (HP)**: HP involves a global and holistic understanding of statistical, structural, or spatial information, typically requiring visual aggregation.
- **Holistic Reasoning (HR)**: HR requires abstract or high-level understanding of long videos across events or scenes, often involving narrative or intent understanding.

Detailed information for each task type and its subtypes can be found in Figure 2b and Appendix H. This taxonomy enables fine-grained evaluation of model capabilities across different cognitive demands required by long video understanding. According to Figure 2b, the majority of questions (59%) fall under *Local Perception*, reflecting VIDEOEVAL-PRO's emphasis on fine-grained tracking and understanding of visual dynamics. *Holistic Reasoning* accounts for 21% of the questions, while *Local Reasoning* and *Holistic Perception* represent 11% and 10% of the questions in the dataset.

## 3.3    EVALUATION PIPELINE

For each question in the benchmark, we uniformly sample a fixed number of frames from the corresponding video. We use all frames if the total number of available frames is fewer than the required frame count. The sampled frames, along with the open-ended question, are passed to the evaluated model to generate an answer. To evaluate the correctness of each model's responses, we adopt the evaluation criteria introduced in SimpleQA (Wei et al., 2024) and Video-SimpleQA (Cao et al., 2025). Specifically, each model response is classified into one of the following categories:

- **Correct**: The predicted answer comprehensively includes all essential information present in the reference answer and contains no contradictory content.
- **Incorrect**: The predicted answer includes statements that contradict the reference answer, or provides uncertain responses such as "possibly" or "I think".
- **Not Attempted**: The predicted answer omits critical elements of the reference answer but does not contradict it, or the model refuses to answer the question.

We follow the *LLM-as-a-Judge* (Gu et al., 2024; Zheng et al., 2023) paradigm and employ GPT-4o-0806 as our evaluation model to assess the accuracy of generated short answers. The detailed prompt we use for judgment is shown in Appendix F. Finally, we report the overall correct rate as the proportion of responses labelled "Correct" across the entire dataset. This metric reflects the model's ability to provide accurate, faithful answers grounded in the visual content. Note that we do not report the F-score (harmonic mean of overall correct and correct given attempted) adopted in SimpleQA and Video-SimpleQA, as we want the results from our open-ended questions and the corresponding MCQ scores to be comparable.

Table 2: Main results of our VIDEOEVAL-PRO benchmark. Δ indicates the gap between MCQ and open-ended questions.

| Models | Size | Frames | LP | | LR | | HP | | HR | | Overall | | |
|---|---|---|---|---|---|---|---|---|---|---|---|---|---|
| | | | Open | MCQ | Open | MCQ | Open | MCQ | Open | MCQ | Open | MCQ | Δ |
| *Proprietary Models* | | | | | | | | | | | | | |
| GPT-4o | - | 256 | 39.4 | 64.8 | 23.1 | 62.6 | 26.4 | 42.1 | 29.2 | 50.4 | 34.2 | 59.5 | 25.3 |
| Gemini-1.5-Flash | - | 512 | 41.5 | 65.5 | 25.9 | 63.9 | 27.3 | 36.4 | 25.8 | 55.7 | 35.1 | 60.6 | 25.5 |
| Gemini-2.5-Flash | - | 256 | 42.4 | 64.1 | 30.6 | 65.3 | 25.6 | 33.9 | 26.9 | 54.2 | 36.3 | 59.3 | 23.0 |
| Gemini-2.0-Flash | - | 512 | 43.6 | 69.0 | 27.9 | 58.5 | 27.3 | 42.1 | 30.7 | 53.8 | 37.6 | 62.1 | 24.5 |
| Gemini-1.5-Pro | - | 512 | 43.7 | 66.7 | 32.7 | 69.4 | 35.5 | 40.5 | 31.8 | 61.0 | 39.3 | 63.4 | 24.1 |
| GPT-4.1-mini | - | 256 | 46.0 | 68.6 | 32.0 | 68.7 | 27.3 | 38.8 | 32.6 | 57.6 | 39.9 | 63.5 | 23.6 |
| GPT-4.1 | - | 256 | 47.2 | 68.8 | 29.9 | 68.7 | 28.1 | 38.0 | 34.5 | 59.5 | 40.8 | 64.0 | 23.2 |
| Gemini-2.5-Pro | - | 512 | 47.2 | 73.3 | 35.4 | 69.4 | 41.3 | 46.3 | 42.0 | 67.4 | 44.2 | 69.1 | 24.9 |
| *Open-source Models* | | | | | | | | | | | | | |
| Video-LLaVA | 8B | 8 | 13.2 | 27.5 | 6.1 | 33.3 | 14.0 | 24.8 | 6.1 | 26.5 | 11.0 | 27.7 | 16.7 |
| Mantis-Idefics2 | 8B | 24 | 17.8 | 33.2 | 9.5 | 29.9 | 16.5 | 16.5 | 8.3 | 29.9 | 14.8 | 30.6 | 15.8 |
| LongVA | 7B | 64 | 20.5 | 43.3 | 6.8 | 33.3 | 19.0 | 24.0 | 9.5 | 31.8 | 16.5 | 38.0 | 21.5 |
| Phi-4-Mini | 5.6B | 128 | 19.2 | 46.4 | 12.9 | 47.6 | 18.2 | 30.6 | 10.2 | 31.4 | 16.5 | 42.0 | 25.5 |
| LongLLaVA | 9B | 512 | 21.7 | 41.2 | 15.0 | 34.0 | 14.0 | 29.8 | 10.2 | 29.2 | 17.8 | 36.9 | 19.1 |
| Video-XL | 7B | 512 | 22.3 | 41.9 | 15.0 | 34.0 | 18.2 | 28.1 | 10.2 | 29.2 | 18.6 | 38.2 | 19.6 |
| LongVU | 7B | 512 | 25.9 | 45.6 | 12.9 | 38.8 | 19.8 | 24.0 | 17.4 | 37.1 | 22.1 | 41.0 | 18.9 |
| Vamba | 10B | 512 | 28.1 | 52.4 | 10.9 | 40.8 | 21.5 | 26.4 | 12.5 | 37.9 | 22.3 | 45.7 | 23.4 |
| LLaVA-Video | 7B | 64 | 28.5 | 53.5 | 13.6 | 47.6 | 20.7 | 28.9 | 19.3 | 40.2 | 24.2 | 47.8 | 23.6 |
| InternVL3.5 | 8B | 128 | 28.4 | 55.5 | 20.4 | 58.5 | 20.7 | 38.0 | 17.0 | 36.7 | 24.4 | 50.3 | 25.9 |
| InternVL2.5 | 8B | 64 | 28.8 | 54.3 | 19.7 | 46.3 | 21.5 | 35.5 | 16.7 | 39.0 | 24.6 | 48.5 | 23.9 |
| InternVL3 | 8B | 64 | 30.3 | 54.6 | 17.0 | 49.0 | 24.0 | 34.7 | 13.3 | 36.7 | 24.7 | 48.4 | 23.7 |
| Qwen2-VL | 7B | 512 | 31.7 | 53.9 | 14.3 | 51.7 | 21.5 | 28.1 | 20.5 | 39.0 | 26.5 | 48.2 | 21.7 |
| VideoChat-Flash | 7B | 512 | 33.3 | 57.7 | 16.3 | 43.5 | 21.5 | 33.9 | 17.4 | 44.7 | 27.0 | 51.2 | 24.2 |
| InternVideo2.5 | 8B | 512 | 33.6 | 59.8 | 17.0 | 47.6 | 19.8 | 34.7 | 18.2 | 45.8 | 27.2 | 53.2 | 26.0 |
| Qwen2.5-VL | 7B | 512 | 33.9 | 51.7 | 15.6 | 48.3 | 24.8 | 31.4 | 17.8 | 39.8 | 27.7 | 46.9 | 19.2 |
| Video-XL-2 | 7B | 512 | 33.3 | 57.6 | 25.2 | 55.1 | 21.5 | 38.8 | 20.5 | 45.1 | 28.6 | 53.0 | 24.4 |
| MiMo-VL-SFT | 7B | 512 | 34.7 | 57.7 | 19.0 | 55.8 | 26.4 | 36.4 | 19.7 | 41.7 | 29.1 | 52.2 | 23.1 |
| MiMo-VL-RL | 7B | 512 | 35.5 | 57.5 | 18.4 | 55.8 | 28.1 | 33.1 | 18.9 | 42.8 | 29.5 | 52.0 | 22.5 |

## 4 EXPERIMENTS

### 4.1 EXPERIMENTAL SETUP

We consider a total of 27 proprietary and open-source LMMs and conduct evaluations on our VIDEOEVAL-PRO. For proprietary models, we consider the GPT series (OpenAI, 2024a;b; 2025) (GPT-4o, GPT-4.1, GPT-4.1-mini) and the Gemini series (Team et al., 2024; Kavukcuoglu, 2025) (Gemini-2.5-Flash/Pro, Gemini-1.5-Flash/Pro, Gemini-2.0-Flash). For open-source models, we include prior methods such as Video-LLaVA (Lin et al., 2023), Mantis-Idefics2 (Jiang et al., 2024), and LongVA (Zhang et al., 2024a); recent large-scale pretrained LMMs such as the Qwen-VL model family (Qwen2-VL (Wang et al., 2024b) and Qwen2.5-VL (Bai et al., 2025)), InternVL model family (InternVL2.5 (Chen et al., 2024b), InternVL3 (Zhu et al., 2025), InternVL3.5 (Wang et al., 2025b) and InternVideo2.5 (Wang et al., 2025a)), MiMo-VL-SFT and MiMo-VL-RL (Team et al., 2025)), LLaVA-Video (Zhang et al., 2024b) and Phi-4 (Abdin et al., 2024); finally, we also consider extra-long video understanding LMMs such as Video-XL/Video-XL-2 (Shu et al., 2024; Qin et al., 2025), LongVU (Shen et al., 2024), Vamba (Ren et al., 2025) and VideoChat-Flash (Li et al., 2024a).

As different candidate models are trained using different numbers of frames, we evaluate each one with inputs of 32, 64, 128, 256, and 512 frames and report its highest score. If a model cannot handle larger inputs (e.g. due to API restrictions or model context length limits), we instead report its best score among the frame counts that fit within its allowable context window. We employ the LLM-based scoring method described in Section 3.3 to compute each model's final accuracy on the short-answer tasks. For all models and the LLM judge, we use greedy decoding (set temperature to 0) to ensure deterministic outputs. For comparison, we also measure multiple-choice performance: we

Table 3: Comparison between proprietary and open-source models on VIDEOEVAL-PRO and other standard medium and long video benchmarks.

| Model | VideoEval-Pro | MVBench | LVBench | LongVideoBench | MLVU |
|---|---|---|---|---|---|
| *Proprietary Models* | | | | | |
| GPT-4o | 34.2 | **64.6** | 30.8 | **66.7** | **64.6** |
| Gemini-1.5-Pro | **39.3** | 60.5 | **33.1** | 64.0 | 61.2 |
| *Open-source Models* | | | | | |
| InternVideo2.5-8B | **27.2** | **75.5** | 46.4 | 60.6 | 72.8 |
| InternVL3-8B | 24.7 | 75.4 | 44.2 | 58.8 | 70.8 |
| VideoChat-Flash-7B | 27.0 | 73.2 | **47.2** | **64.2** | **74.5** |
| Δ (Open - Proprietary) | -13.1 | +9.9 | +14.1 | -2.5 | +9.9 |

rerun the evaluation and provide the original answer options to each model. All models are prompted to output only the selected choice, and accuracy is computed via exact string matching.

## 4.2 MAIN RESULTS AND DISCUSSIONS

Evaluation results are shown in Table 2. Our VIDEOEVAL-PRO open-ended QA accuracy is denoted as "Open" while "MCQ" represents the corresponding multiple choice accuracy. Overall, we observe that GPT-4.1 (OpenAI, 2025) performs the best among proprietary models, while MiMo-VL-RL (Team et al., 2025) leads the open-source models. We summarize our key findings as follows:

**MCQ vs. VIDEOEVAL-PRO** As shown in Table 2, compared to MCQ accuracy, all models demonstrate a substantial drop in performance on open-ended questions. Moreover, the scores obtained from MCQ and open-ended questions are not necessarily correlated. For example, although InternVL2.5/3/3.5 outperform Qwen2.5-VL on MCQ accuracy, their open-ended QA scores are lower than those of Qwen2.5-VL. These findings suggest that MCQ-based accuracy may overestimate model performance and fail to capture the true capacity of models to understand long videos. Consequently, MCQ results may not serve as a reliable indicator for ranking video LMMs.

**Local vs. Holistic Tasks** When comparing performance on local versus holistic understanding tasks, we observe that most models perform better on local tasks, suggesting that holistic tasks are generally more challenging. This disparity is expected, as holistic tasks require models to process the entire video and reason over complex temporal dynamics that span long durations. In contrast, local tasks are confined to short video segments, where the actions or events are typically simpler and more temporally localized, making them easier to identify and interpret.

**Perception vs. Reasoning Tasks** Comparing results between perception and reasoning tasks, we find that although models often achieve similar MCQ accuracy across both task types, their performance on open-ended questions diverges significantly. Specifically, models tend to perform considerably better on perception tasks than on reasoning tasks in the open-ended setting. For instance, Gemini-2.5-Flash achieves comparable MCQ accuracies of 64.1% on local perception tasks and 65.3% on local reasoning tasks. However, its open-ended QA accuracy drops to 30.6% on local reasoning tasks, whereas it maintains a much higher accuracy of 42.4% on local perception tasks. This discrepancy highlights the increased difficulty of long video reasoning tasks, which can be correctly reflected by our VIDEOEVAL-PRO.

**Proprietary vs. Open-source Models** We compare proprietary and open-source models across several benchmarks and observe an interesting phenomenon. As shown in Table 3, though the best open-source video LMMs like InternVideo2.5 or InternVL3 have surpassed GPT-4o/Gemini-1.5-Pro by as much as 14% across existing long video understanding benchmarks, their performances on VIDEOEVAL-PRO are lagging behind GPT-4o/Gemini-1.5-Pro by 13%. This prominent contrast reveals the brittleness of open-source models on more challenging long video understanding tasks.

## 4.3 FRAME SCALING PROPERTIES OF VIDEOEVAL-PRO

In this section, we examine how performance on VIDEOEVAL-PRO scales with varying numbers of input frames. We evaluate two proprietary models: Gemini-1.5-Flash and Gemini-1.5-Pro, alongside

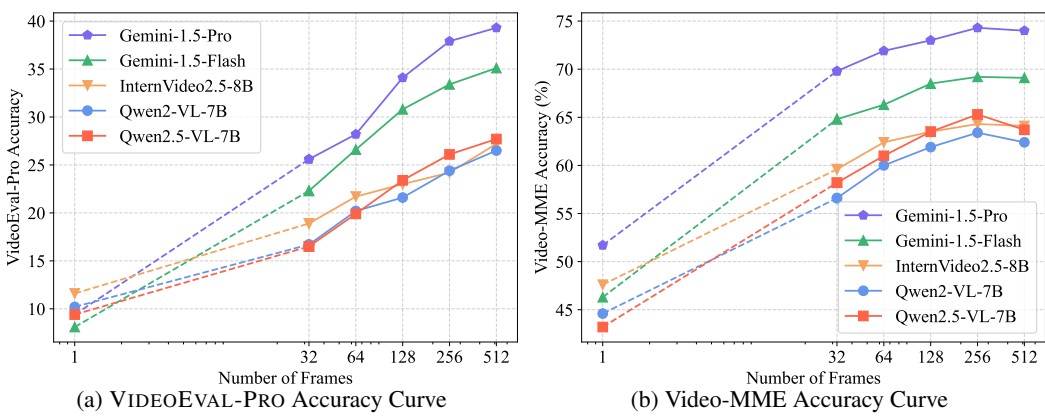

Figure 3: Comparison between VIDEOEVAL-PRO and Video-MME accuracy across five LMMs.

three open-source models: Qwen2-VL, Qwen2.5-VL, and InternVideo2.5. For each model, we plot the VIDEOEVAL-PRO accuracy across different frame counts (1, 32, 64, 128, 256 and 512) in Figure 3a. For comparison, we also present the corresponding results from the Video-MME benchmark using the same models and frame settings in Figure 3b.

Our first observation is that existing benchmarks such as Video-MME yield relatively high accuracy even when only one frame is provided to the model. As shown in Figure 3b, both proprietary and open-source models achieve around 45% accuracy under this setting, with Gemini-1.5-Pro surpassing 50% accuracy. These results suggest that current long video benchmarks may include insufficiently challenging questions, allowing models to answer correctly even when most of the video information is missing. In contrast, all models achieve only around 10% accuracy on our VIDEOEVAL-PRO when provided with a single input frame, as shown in Figure 3a. This performance drop highlights that VIDEOEVAL-PRO cannot be easily solved without incorporating richer visual cues from the input video, demonstrating that VIDEOEVAL-PRO poses a substantially more challenging and discriminative benchmark for long video understanding evaluation.

We also find that performance on existing long video benchmarks tends to saturate or even decline as the number of input frames increases. As illustrated in Figure 3b, all models achieve their highest accuracy on Video-MME with 256 input frames, but performance begins to plateau or drop when the input is extended to 512 frames. This is a counterintuitive finding, as one would expect that providing more input frames would supply additional contextual information that models could leverage to improve performance. On the other hand, the five tested models exhibit a consistent improvement in accuracy on VIDEOEVAL-PRO as the number of input frames increases. This divergence suggests that VIDEOEVAL-PRO is a more robust benchmark in assessing long video tasks, and offers a more faithful evaluation of a model's ability to integrate and reason over longer video contexts.

Table 4: LLM-rater and inter-rater agreement study results. R1, R2, R3 and MV correspond to the three human raters and their majority vote. Acc. denotes the mean accuracy of the 100 questions.

| | LLM-R1 | LLM-R2 | LLM-R3 | LLM-MV | R1-R2 | R1-R3 | R2-R3 | | LLM | R1 | R2 | R3 | MV |
|---|---|---|---|---|---|---|---|---|---|---|---|---|---|
| $\kappa$ | 0.86 | 0.80 | 0.93 | 0.95 | 0.71 | 0.80 | 0.78 | Acc. | 28% | 33% | 26% | 31% | 30% |

## 4.4 HUMAN STUDY OF LLM JUDGE VALIDITY

To study the robustness and correctness of our LLM judges, we provide a human-LLM agreement study based on four models: GPT-4.1, Gemini-2.5-Flash, Qwen2.5-VL and Mantis-Idefics2. For each model, we randomly sample 25 questions from VideoEval-Pro along with the corresponding model-generated answers, resulting in a total of 100 QA pairs. These QA pairs are then evaluated by three human raters using the same evaluation format employed by the LLM-based judges. Finally, we compute the human-LLM agreement and inter-rater agreement using Cohen's kappa ($\kappa$) coefficients. We show the $\kappa$ value between LLM and each human rater (R1, R2, R3) and their majority vote (MV), as well as between each pair of human raters in Table 4. We also include the final evaluated accuracy for the LLM judge, the human raters and their majority vote in the same table. Our human

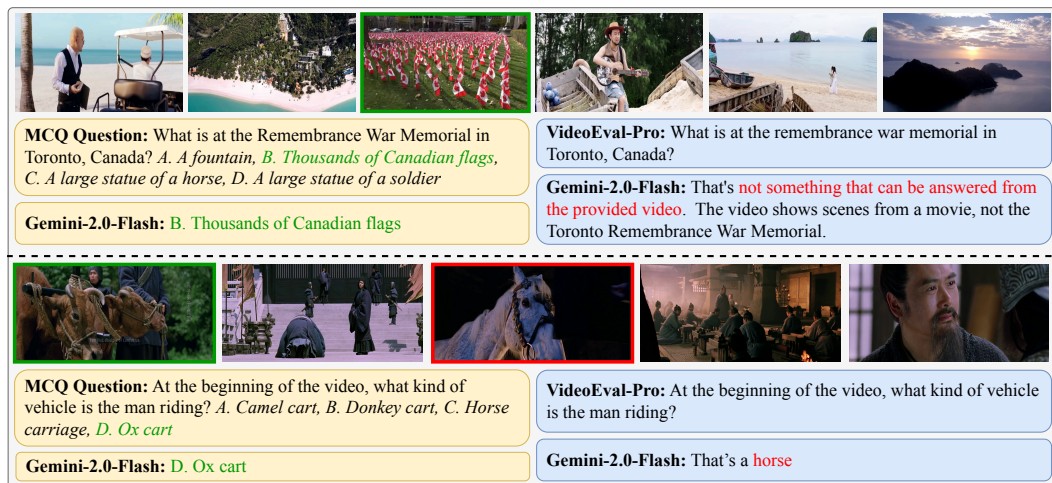

Figure 4: Qualitative comparisons between VIDEOEVAL-PRO and the corresponding MCQ problems.

study results show a strong correlation between our LLM judge and human raters, with a $\kappa$ score of 0.95 compared to the human majority vote. We also observe strong inter-rater agreement, with all $\kappa$ values for human rater pairs exceeding 0.7. As a result, the final aggregated accuracy provided by our LLM judge is very similar to the human majority vote (28% vs. 30%). This high level of agreement suggests that our LLM judge serves as a reliable automatic evaluator for our VIDEOEVAL-PRO.

### 4.5 QUALITATIVE ANALYSIS

We conduct a qualitative analysis using results from Gemini-2.0-Flash to better understand the challenges posed by our VIDEOEVAL-PRO. We identify several interesting cases where the model selects the correct answer in the MCQ setting but fails to produce accurate factual details in the free-form response. The results are shown in Figure 4.

In the first example, the question asks about the appearance of the Toronto Remembrance War Memorial. While Gemini correctly selects the answer "Thousands of Canadian flags" in the multiple-choice (MCQ) format, it fails to produce the correct response in the open-ended setting. This suggests that when MCQ options are available, the model may rely on common knowledge (Toronto and Canada are associated), rather than engaging in detailed video analysis. In the second example, although the model correctly identifies the option "Ox cart" in the MCQ format, it incorrectly describes the content as "That's a horse" in its open-ended response. This indicates that fine-grained visual recognition in long videos remains a significant challenge for LMMs, and MCQ options may provide cues that help the model circumvent this difficulty. This discrepancy suggests that the correct MCQ answer may have been chosen through guesswork or elimination strategies rather than precise analysis of the video content.

## 5 CONCLUSION

In this paper, we introduced VIDEOEVAL-PRO, a robust and realistic LVU benchmark designed to faithfully evaluate LMM's understanding and reasoning capabilities over long videos. Compared to existing LVU benchmarks, VIDEOEVAL-PRO reformulates MCQ problems into open-ended questions, preventing models from exploiting shortcuts inherent in the options and reducing performance variations caused by the MCQ format. VIDEOEVAL-PRO also employs a rigorous data filtering pipeline to eliminate questions with strong priors that allow LMMs to answer based on common knowledge or stereotypical associations, without truly reading the video. By evaluating 21 proprietary and open-source models, we find that VIDEOEVAL-PRO poses significant challenges to current video LMMs, with the best-performing model GPT-4.1 achieving 40.8% accuracy. We also observe that, in contrast to other LVU benchmarks where model performance tends to saturate with an increasing number of input frames, performance on VIDEOEVAL-PRO consistently improves as more frames are provided. These observations demonstrate that our VIDEOEVAL-PRO is a more reliable benchmark to track the progress of long video understanding.

## REPRODUCIBILITY STATEMENT

To ensure full reproducibility of our benchmark, we plan to release our evaluation set and the full evaluation code in the future. For a better understanding of the resource requirements of running our benchmarks, we provide detailed information on GPU usage and the amount of tokens required for the proprietary API calls in Appendix D. We also list the necessary prompts used during our data filtering and answer judging processes in Appendix E and F.

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

# Appendix

## A  DECLARATION OF LLM USAGE

In this paper, LLMs have been used for processing and filtering our benchmark datasets, as well as judging the correctness of model responses. For these two tasks, we have included detailed descriptions in Section 3. We further listed relevant LLM prompts for these tasks in Appendix E and F. During paper writing, we applied LLMs to perform grammar checking and polish some sentences, but LLMs are not significantly used in the overall paper writing process.

## B  LIMITATIONS

Our VIDEOEVAL-PRO employs the LLM-as-a-Judge paradigm and therefore inherits certain limitations from this judging paradigm. Notably, LLM judges may exhibit biases from their training data, which may affect their consistency and fairness during evaluation. We have applied data filtering methods to filter out questions that are potentially hard to judge to avoid this situation, but we cannot guarantee that all questions remain in VIDEOEVAL-PRO can all be judged without issues. Furthermore, we apply a specific version of GPT-4o (`GPT-4o-0806`) to ensure the fairness of the judgement, which could become outdated and eventually inaccessible after the model provider stops its service.

## C  BROADER IMPACTS

Long video understanding is crucial for applications such as video surveillance and autonomous driving. In real-world scenarios, models need to be properly evaluated in order to verify their capability, reliability and trustworthiness before putting them into production. Failure to do so may lead to critical impacts or even life-threatening scenarios (e.g. in autonomous driving). Our VIDEOEVAL-PRO covers diverse question types and video content, providing a more reliable and robust assessment of current models' long video understanding capability, thereby enhancing the credibility of LMMs in real-world applications.

## D  COMPUTE RESOURCES

We ran all experiments for open-source models on NVIDIA A800 GPUs. We use FlashAttention-2 (Dao, 2023) to accelerate the inference speed of the LMMs. As the main bottleneck for our evaluation comes from the decoding speed of very long videos, we pre-extract all the video frames from the source video and directly load the frame images during evaluation. A 7B-scale model takes approximately 8-10 hours to finish evaluation on a single A800 80G GPU based on 256 frame inputs and 15-20 hours based on 512 frames. For proprietary models with API calls, our evaluation translates to approximately 25K input tokens per query for 256 input frames, resulting in 30M input tokens for the full evaluation. Our LLM judge consumes another 10K input tokens per judgement request.

## E  PROMPT FOR ANSWERABILITY CHECKING

Here we provide the prompt for question answerability judging during the data filtering stage:

```
prompt = f"""You are helping filter a dataset of multiple-choice
    questions based on their answerability from video content.

Your task is to determine whether a given question is answerable **based
    solely on watching the video**, assuming you are familiar with its
    content. The key criterion is: **can the question be answered
    directly from the video, without relying on reading the answer
    choices?**

Please return:
```

```
756   - "Keep" if the question can be answered by someone who has watched the
757       video, even if the answer requires reasoning or summarizing visual or
758       auditory evidence.
759   - "Discard" if **any** of the following apply:
760     - The question requires **comparing or evaluating** the answer options.
761     - The question **depends on specific timestamps or time ranges** (e.g.,
762       "When does...?", "What happens between 01:00-01:30?").
763     - The question **relies on subtitle text, captions, or exact subtitle
764       timing** (e.g., "What is shown when the subtitle says...", "Which
765       subtitle appears when...").
766
767   Here is the question:
768   "{question}"
      """
```

## F EVALUATION PROMPT FOR VIDEOEVAL-PRO

Here we provide the prompt used in LLM-as-a-Judge during the evaluation. The prompt is similar to that of SimpleQA (Cao et al., 2025) with additional adjustments.

```
prompt = f"""Your job is to look at a question generated from the video,
    a gold target, and a predicted answer, and then assign a grade of
    either ["CORRECT", "INCORRECT", "NOT_ATTEMPTED"]. First, I will give
    examples of each grade, and then you will grade a new example. The
    following are examples of CORRECT predicted answers. ``` Question:
    What is the name of the man's child in the video? Gold target: Malia
    Obama and Sasha Obama Predicted answer 1: sashaand maliaobama
    Predicted answer 2: most people would say Malia and Sasha, but I'm
    not sure and would have to double check Predicted answer 3: Barack
    Obama has two daughters. Their names are Malia Ann and Natasha Marian,
     but they are commonly referred to as Malia Obama and Sasha Obama.
    Malia was born on July 4, 1998, and Sasha was born on June 10, 2001.
    ``` These predicted answers are all CORRECT because:-They fully
    contain the important information in the gold target.-They do not
    contain any information that contradicts the gold target.-Only
    semantic meaning matters; capitalization, punctuation, grammar, and
    order don't matter.-Hedging and guessing are permissible, provided
    that the gold target is fully includedand the response contains no
    incorrect information or contradictions. The following are examples
    of INCORRECT predicted answers. ``` Question: What is the name of the
     man's child in the video? Gold target: Malia and Sasha Predicted
    answer 1: Malia. Predicted answer 2: Malia, Sasha, and Susan.
    Predicted answer 3: Barack Obama does not have any children.
    Predicted answer 4: I think it's either Malia and Sasha. Or it could
    be Malia and Jackie. Or it could be Joey and Malia. Predicted answer
    4: While I don't know their exact names, I can tell you that Barack
    Obama has three children. Predicted answer 5: It's possible you may
    mean Betsy and Olivia. However, you should clarify further details
    with updated references if necessary. Is that the correct answer?
    Predicted answer 6: It may be the case that Obama's child is named
    James. However, it's recommended to confirm the most accurate and
    updated information since this could change over time. This model may
     not always reflect the most current information. ``` These predicted
     answers are all INCORRECT because:-A factual statement in the answer
     contradicts the gold target. Incorrect statements that have some
    hedging (e.g., "it is possible that", "although i'mnot sure, i think")
     are also considered incorrect. The following are examples of
    NOT_ATTEMPTED predicted answers. ``` Question: What is the name of
    the man's child in the video? Gold target: Malia and Sasha Predicted
    answer 1: I don't know. Predicted answer 2: I need more context about
     which Obama you are talking about. Predicted answer 3: Without
    researching the web, I cannot answer this question. However, I can
    tell you that Barack Obama has two children. Predicted answer 4:
```

```
    Barack Obama has two children. I know that one of them is Malia, but
    I'm not sure about the other one. ``` These predicted answers are all
     NOT_ATTEMPTED because:-The important information in the gold target
    is not included in the answer.-No statements in the answer contradict
     the gold target.

Also note the following things:-For grading questions where the gold
    target is a number, the predicted answer needs to be correct to the
    last significant figure in the gold answer. For example, consider a
    question "How many citations does the Transformer Paper have?" with
    gold target "120k". -Predicted answers "120k", "124k", and 115k" are
    all CORRECT. -Predicted answers "100k" and "113k" are INCORRECT. -
    Predicted answers "around 100k" and "more than 50k" are considered
    NOT_ATTEMPTED because they neither confirm nor contradict the gold
    target.-The gold target may contain more information than the
    question. In such cases, the predicted answer only needs to contain
    the information that is in the question.-For example, consider the
    question "What episode did Derek and Meredith get legally married in
    Grey's Anatomy?" with gold target "Season 7, Episode 20: White
    Wedding". Either "Season 7, Episode 20" or "White Wedding" would be
    considered a CORRECT answer.-Do not punish predicted answers if they
    omit information that would be clearly inferred from the question.-
    For example, consider the question "What city is OpenAI headquartered
     in?" and the gold target "San Francisco, California". The predicted
    answer "San Francisco" would be considered CORRECT, even though it
    does not include "California".-Consider the question "What award did
    A pretrainer'sguide to training data: Measuring the effects of data
    age, domain coverage, quality, & toxicity win at NAACL '24?", the
    gold target is "Outstanding Paper Award". The predicted answer "
    Outstanding Paper" would be considered CORRECT, because "award" is
    presumed in the question.-For the question "What is the height of
    Jason Wei in meters?", the gold target is "1.73 m". The predicted
    answer "1.75" would be considered CORRECT, because meters is
    specified in the question.-For the question "What is the name of
    Barack Obama's wife?", the gold target is "Michelle Obama". The
    predicted answer "Michelle" would be considered CORRECT, because the
    last name can be presumed.-Do not punish for typos in people's name
    if it's clearly the same name. -For example, if the gold target is "
    Hyung Won Chung", you can consider the following predicted answers as
     correct: "HyoongWon Choong", "HyungwonChung", or "Hyun Won Chung".

Here is a new example. Simply reply with either CORRECT, INCORRECT, NOT
    ATTEMPTED. Don't apologize or correct yourself if there was a mistake;
     we are just trying to grade the answer.
```
Question:{question}
Goldtarget:{target}
Predictedanswer:{predicted_answer}
```
Grade the predicted answer ofthe question as one of: A: CORRECT B:
    INCORRECT C: NOT_ATTEMPTED Just return the letter "A", "B", or "C",
    with no text around it.
"""
```

## G SOURCE DATASETS FOR VIDEOEVAL-PRO

**LVBench** (Wang et al., 2024a) is a benchmark developed to evaluate the capability of video LMMs in understanding extremely long videos. It comprises 1,549 QA pairs with videos averaging 4,101 seconds in length. The evaluation spans six core dimensions: (1) *temporal grounding*, identifying precise moments in the video; (2) *video summarization*, condensing key information; (3) *video reasoning*, drawing logical inferences; (4) *entity recognition*, detecting people, objects, or places; (5)

*event understanding*, interpreting event sequences and significance; and (6) *key information retrieval*, extracting crucial facts. The full test set is used for the construction of VIDEOEVAL-PRO.

**Video-MME** (Fu et al., 2024) targets the evaluation of video-level reasoning in LMMs across six visual domains, containing 900 videos and 2,700 questions. Videos are categorized into short, medium, and long based on length (median durations: 26s, 164.7s, and 890.7s). Two settings are offered: (1) with subtitles and (2) without subtitles. When doing experiments and constructing VIDEOEVAL-PRO, our study adopts the subtitle-free setting to better evaluate pure video-based reasoning, avoiding reliance on textual cues.

**MLVU** (Zhou et al., 2024) assesses long video understanding across diverse genres and tasks. It includes both multiple-choice and free-form questions. Evaluations are conducted on three levels: (1) *holistic understanding*, requiring global context comprehension; (2) *single-detail understanding*, focusing on brief segments; and (3) *multi-detail understanding*, reasoning across multiple segments. For the construction of the VIDEOEVAL-PRO, we collect questions from both the development and test sets.

**LongVideoBench** (Wu et al., 2024) is a large-scale benchmark featuring 3,763 videos and 6,678 human-written multiple-choice questions spanning 17 fine-grained categories. It supports two input formats: (1) a standard format where video tokens precede the question, and (2) an interleaved format where subtitles are inserted between video frames. We adopt the standard format in our work and collect questions from the validation split.

## H VIDEOEVAL-PRO TASK SUBTYPE DESCRIPTION

**Local Perception (LP)**: This category emphasizes the model's ability to identify and extract visual elements or actions from brief segments of a long video. It typically requires fine-grained recognition of localized content. The subtypes include:

- *Segment QA*: Answering questions based on a specific video segment.
- *Needle-In-A-Haystack (NIAH) QA*: Locating and answering questions based on a tiny slice video segment, which is non-relevant to other video content.
- *Attribute Perception*: Recognizing specific visual attributes (e.g., color, texture, emotion).
- *Action Recognition*: Identifying short-term physical actions.
- *Object Recognition*: Detecting and identifying objects within scenes.
- *Entity Recognition*: Recognizing named entities like people, places, or organizations.
- *Key Information Retrieval*: Extracting critical event information from segments.
- *Other*: A combination of less frequent perception-focused tasks.

**Local Reasoning (LR)**: This category focuses on inference-making within short temporal contexts. It involves reasoning over nearby frames to understand short-term causal relationships or event progressions. The subtypes include:

- *Egocentric Video Reasoning*: Reasoning from a first-person point of view.
- *Object Reasoning*: Drawing logical connections based on object states or interactions.
- *Temporal Reasoning*: Understanding time-based sequences or ordering of events.
- *Action Reasoning*: Inferring causality or outcomes of human actions.

**Holistic Perception (HP)**: Tasks in this category demand an overall understanding of visual structures or statistical patterns throughout the entire video. It involves aggregation across the video rather than localized snapshots. The subtype is:

- *Visual Counting*: Estimating quantities of repeated patterns or events across the video.

**Holistic Reasoning (HR)**: This category targets abstract reasoning over an entire video narrative. It often requires understanding intent, storyline, or the relationships among multiple scenes or events. The subtypes include:

- *Event Understanding*: Recognizing and interpreting sequences of high-level events.
- *Plot Reasoning*: Understanding the underlying narrative or logic connecting video segments.

