# OpenReview forum: "VideoEval-Pro: Robust and Realistic Long Video Understanding Evaluation"
_ICLR.cc/2026/Conference — ICLR 2026 Conference Withdrawn Submission_

### Official Review · Reviewer_tHDe · 2025-10-28

**Soundness:** 3
**Presentation:** 3
**Contribution:** 3
**Rating:** 6
**Confidence:** 5

**Summary:**

The paper introduces VideoEval-Pro, a benchmark aimed at assessing how well multimodal large language models (MLLMs) understand long-form videos under realistic and open-ended conditions. The authors argue that existing multiple-choice-based video benchmarks overestimate performance by allowing models to exploit statistical priors or scene biases, rather than engaging in genuine temporal reasoning. To remedy this, they transform several existing datasets (including MLVU, LVBench, LongVideoBench, and Video-MME) into an open-ended QA format, then implement a multi-stage filtering pipeline that removes trivially solvable or short-context items.

The resulting benchmark comprises roughly 1,300 question–answer pairs across 465 long videos (averaging about 40 minutes each). Each item is labeled by both spatial scope (local vs holistic) and cognitive demand (perception vs reasoning), yielding a structured taxonomy that enables finer-grained evaluation. Responses from 27 models spanning proprietary and open-source MLLMs, are scored automatically by a GPT-4o-based evaluator, validated against human ratings on a held-out sample (showing strong agreement, κ≈0.95).

Empirically, the authors find that open-ended evaluation produces substantially lower absolute scores and different model rankings than the traditional MCQ setup, implying that multiple-choice testing inflates apparent capability. Frame-scaling experiments (1–512 frames) confirm that VideoEval-Pro demands genuine temporal integration, as performance improves consistently with longer visual context: contrary to saturation observed in prior datasets. Notably, commercial models outperform open-source ones under this open-ended regime, suggesting that true long-range understanding remains unsolved. The authors frame VideoEval-Pro as a next-generation diagnostic tool for long video comprehension, with a commitment to public release of data, prompts, and scripts.

**Strengths:**

1) By converting items from four established LVU sets into open-ended, short-answer prompts and filtering out single-frame-solvable questions, the benchmark squarely tests temporal evidence use rather than option-elimination heuristics. The head-to-head comparison (same items: MCQ vs open-ended) shows >25-point drops and rank inversions, which provides great insights.

2) The multi-stage curation (remove short videos, drop low-answerability and prior-driven items) produces a set of 1,289 QA over 465 videos (mean ~38.25 min), and the taxonomy spans Local/Holistic × Perception/Reasoning, which supports structured analyses across task families.

3) Unlike earlier MCQ benchmarks where accuracy can saturate or even degrade with more frames, VideoEval-Pro shows monotonic gains as frames increase (1→128→256→512), empirically indicating that items actually require long-range integration.

4) The table contrasting proprietary vs open-source models across VideoEval-Pro and MCQ suites (LVBench/MLVU/etc.) demonstrates cleartly that the evaluation format can flip perceived leadership—an important community-level takeaway.

**Weaknesses:**

1)Section 3.3 of the paper states that frames are “uniformly sampled” at a fixed count for each model evaluation, but it does not analyze whether this sampling strategy is optimal or fair across heterogeneous video types. for eg.
Action-heavy videos might require denser temporal sampling to capture relevant cues, while dialogue-heavy or static-scene videos might be well represented by sparser frames emphasizing semantic rather than motion information. The paper’s results (e.g., the monotonic frame-scaling curve) show that more frames help on average, but the absence of per-genre or per-content sampling sensitivity analysis leaves open whether uniform sampling under- or over-represents some content types. The authors could strengthen the work by providing a small ablation comparing uniform vs. motion- or scene-aware sampling across genres or providing a justification for the same.

2) The human–LLM agreement (κ≈0.95) is based on 100 items total; there’s no per-slice reliability by Local/Holistic or Perception/Reasoning, nor confidence intervals. Given the 15 sub-types and the model mix, the current sample seem underpowered to guarantee uniform judge fidelity.

3) The benchmark seems to restrict answer complexity, for instance, Table 1 reports a mean answer length of roughly 2 words, and Section 3.2 notes that questions with answers exceeding five words are filtered out. While this design choice enhances the reliability of judging automatically, it perhaps also narrows the expressive range of the evaluation. As a result, VideoEval-Pro primarily tests short factual retrieval rather than the compositional or explanatory reasoning that is central to long-video understanding. The qualitative samples in Figure 4, where both references and predictions are brief noun or verb phrases, reinforce this limitation. Incorporating a subset with longer, multi-clause answers, even at some cost in scoring stability would help to better evaluate and probe longer narrative and causal reasoning abilities.

4) Although VideoEval-Pro draws its questions from existing MCQ benchmarks, the short-answer filtering (≤5 words) introduced during conversion, may be to improves automatic judging reliability, but it perhaps also compresses the expressive range of responses i.e. potentially under-representing the multi-step, compositional reasoning that long-form video understanding should evaluate. The qualitative samples in Figure 4, where both references and predictions are brief noun or verb phrases, seem to corroborate the limitation of this design choice.

5) Because all items originate from four MCQ sources, residual topic/genre skews can perhaps persist even after filtering. The paper notes the source mix (e.g., LVBench dominance in the pool) but does not provide per-source/genre breakdowns for open-ended accuracy or frame-scaling curves.

**Questions:**

1) Can you provide qualitative analysis of systematic failure cases beyond Figure 4’s two examples, for instance, do failure modes differ on Local vs. Holistic tasks when comparing MCQ and open-ended?

2) It would make the paper stronger if the authors can consider re-running the answerability and single-frame filters with a second, diverse judge (e.g., GPT-4o or an open model) and retaining items on which judges disagree? For instance, a short “agreement matrix” would quantify curation robustness.

3) A per-source (Video-MME vs. LVBench vs. LongVideoBench vs. MLVU) and per-genre breakdown of open-ended accuracy and frame-scaling behavior would help reveal any residual inheritance effects from the parent datasets and clarify whether certain content domains are systematically easier or harder. This addition would help to strengthen the benchmark.

4) The paper reports an overall human–LLM agreement of κ≈0.95 on 100 samples, but does not provide confidence intervals or task-wise reliability. If the authors could report κ with 95% confidence intervals for each of the main task axes (Local vs. Holistic, Perception vs. Reasoning), or, if feasible, consider conducting a  slightly larger (~400-item) stratified audit to confirm that the automatic judge performs consistently across task categories.

---

### Official Review · Reviewer_i2b6 · 2025-10-31

**Soundness:** 2
**Presentation:** 3
**Contribution:** 2
**Rating:** 2
**Confidence:** 3

**Summary:**

The paper proposes VIDEOEVAL-PRO, a realistic LVU benchmark containing questions with open-ended short-answer. The authors describe limitations of existing video understanding benchmarks as:
1. MCQ-based benchmarks’ evaluation results are inflated due to the possibility of guessing the correct answer
2. Significant portion of questions in current benchmarks have strong priors to allow models to answer directly without even reading the input video
3. increasing the number of frames does not necessarily lead to improvement on existing benchmarks
The authors claim that VIDEOEVAL-PRO provides better assessment of true video understanding skill and since it has open-ended answers it is not prone to inflated performance by guessing, etc.

**Strengths:**

1. The paper is well motivated and written. Especially the problem of open-ended generation v/s MCQ-based evaluation is a timely and interesting one to study.
2. The paper conducts comprehensive core evaluation experiments with many different open-source and proprietary models.

**Weaknesses:**

While I like the motivation, I think the paper in its current form falls short in establishing the core usefulness of their proposed benchmark as well as improving our understanding of why models struggle with open-ended generation v/s MCQ-based evaluation. Please see the Questions for more weaknesses, details and specific questions.

**Questions:**

1. The paper claims to evaluate more realistic and reliable long video understanding by ensuring that all the videos selected are greater than some duration and analyzed frame rate scaling shows an increasing trend. However, that is insufficient since many of the accuracy improvements in high frame-rate regime could simply be due to short-span reasoning questions for which the underlying frame information was not available at all. For a stronger claim for reliable long video understanding evaluation an analysis with temporal certificates as described in [1] is required. Could the authors perform this and establish their temporal certificates are in line with their expected improvements with increasing frames.

2. Another consideration which makes temporal certificates important is that the question answerability and difficulty criteria seems to be entirely determined by heuristics based on a single model i.e., Gemini-2.0-Flash. How can we ensure that the answerability is actually lacking v/s Gemini-2.0-Flash simply making a mistake. Could the author repeat this analysis with an ensemble of other high-performing models to minimize bias? Ideally, an human evaluation here would be great since the number of videos / QA pairs are not terribly large.

3. How are the questions binned into the different task types and categories?

4. There seems to be missing related work discussion, as [2] also show that models' open-ended responses can widely differ (and generally underperform) MCQ-based evaluation. Could the authors establish how their work differs in terms of the overarching conclusions regarding open-ended v/s closed-form QA?

5. I didn’t find the qualitative examples discussed in 4.5 very convincing. For instance, for example two the authors talk about how the model correctly identifies an Ox cart in one case but incorrectly says horse in another. However, since there is a horse related option in the MCQ as well, and the video content is same in both, this seems more of a chance occurring. For instance, does the model always answer “horse” in an open-ended setting. Since open-ended v/s MCQ-based is one of the core takeaways of the work, there needs to  be a deeper analysis of different failure modes that are unique to open-ended generation, and perhaps quantifying them across the benchmark for different evaluation models.


[1] EgoSchema: A Diagnostic Benchmark for Very Long-form Video Language Understanding

[2] ARGUS: Hallucination and Omission Evaluation in Video-LLMs

---

### Official Review · Reviewer_Ypnc · 2025-10-31

**Soundness:** 2
**Presentation:** 2
**Contribution:** 2
**Rating:** 2
**Confidence:** 5

**Summary:**

This paper proposes a lightweight video benchmark that contains open-ended short-answer questions. The benchmark focuses on robustness and realism, which the authors argue are often overlooked in existing benchmarks. The work identifies several issues in current benchmarks and re-filters them according to specific rules, then reformats the questions into an open-ended style to build a new benchmark for more comprehensive evaluation of video understanding models.

**Strengths:**

The motivation to filter out existing benchmarks is good, as is the use of open-ended questions instead of multiple-choice ones. However, I’m sorry to say that I really cannot find enough strengths in this paper.

**Weaknesses:**

First, the two key points of this benchmark—robustness and realism—are not clearly justified in the paper. For example, it is unclear why this benchmark is considered more robust. Is it because of the performance drop caused by changing multiple-choice questions (MCQs) into open-ended ones?

Second, the main diagram only shows that “VIDEOEVAL-PRO cannot be effectively solved with a single input frame, and performance scales consistently with more frames.” However, the authors should compare their results with more benchmarks, rather than only with VideoMME.

Third, this benchmark does not bring new insights to the video understanding community. It mainly recollects existing benchmarks and filters them based on certain rules (e.g., one-frame answerability), then reformats them into open-ended questions.

**Questions:**

Q1: In the related work section, the authors only mention the difference between MCQ and open-ended formats. I would like to see how VideoEval-Pro truly differs from other benchmarks such as VideoMME, LVBench, LongVideoBench, and others.
Q2: Why filter out questions like Timestamp-dependent questions, which are very important for robust video understanding.


Suggestions: Benchmarks should contribute in terms of the dataset itself, the evaluation metrics, and the insights they provide — none of which are clearly demonstrated in this paper. I suggest the authors think more deeply about these aspects.

---

### Official Review · Reviewer_WqMH · 2025-11-11

**Soundness:** 2
**Presentation:** 2
**Contribution:** 2
**Rating:** 4
**Confidence:** 4

**Summary:**

The paper introduces VIDEOEVAL-PRO, an open-ended, short-answer benchmark for long video understanding (LVU). It starts from 5,562 MCQs drawn from four LVU datasets (Video-MME, MLVU, LVBench, LongVideoBench), converts them to free-form QAs, and applies duration, answer-length, answerability, and difficulty filters to reduce priors/guessability. The final set has 1,289 Q&As over 465 videos (avg 38.25 min). Evaluation uses uniform frame sampling at varying budgets (1–512 frames), LLM-as-a-judge (GPT-4o-0806) with a Correct/Incorrect/Not-Attempted rubric, reporting the overall Correct rate. Across 27 models, the authors find: (i) >25% drops from MCQ to open-ended; (ii) weak correlation between MCQ and open-ended rankings; (iii) local > holistic and perception > reasoning; (iv) proprietary models regain a lead over open-source on this harder setting; and (v) monotonic gains with more frames on VIDEOEVAL-PRO (vs. saturation/declines on MCQ benchmarks), and ~10% single-frame accuracy, indicating real temporal reliance. A human study reports κ = 0.95 agreement between the LLM judge and the human majority.

**Strengths:**

•	Clear problem diagnosis. MCQ guessability + strong priors can inflate LVU scores; the benchmark explicitly targets these issues.

•	Open-ended reformulation. Short answers (avg 2.1 words) reduce option-structure shortcuts and force models to rely on video content.

•	Thoughtful filtering. Multi-stage filters (duration ≥10 min, concise answers, answerability checks, one-frame difficulty screen) aim to remove trivial/option-dependent items.

•	Task taxonomy. Separation into Local vs Holistic and Perception vs Reasoning enables targeted analysis of failure modes.

•	Frame-scaling analysis. Monotonic improvements with more frames suggest the benchmark actually rewards temporal integration.

**Weaknesses:**

1.	**Limited novelty in data:**
The work does not introduce new videos or fresh human annotations; it repackages prior benchmarks by dropping distractors and keeping the correct option as the gold. The motivation for “MCQ → open-ended” is insufficiently argued: beyond showing a drop in accuracy, the paper does not establish that the open-ended format (with 2-word answers and an LLM judge) is a more faithful measure of LVU rather than a stricter or noisier one. A clearer goal/assumption–evidence link is needed (e.g., human study by task type/length beyond 100 items, inter-judge ablations, synonym handling).

2.	**Filtration under-targets visual reliance:**
The pipeline mainly removes short videos and long textual answers; it also screens for answerability/difficulty, but these steps rely on text-only/LLM heuristics. There is no explicit “vision-first” or “world-knowledge” filter (e.g., question-only controls, numeral masking, named-entity paraphrases, or leakage audits) that proves the remaining items truly require video evidence rather than priors. As written, the benchmark may still inherit textual/style biases from source datasets.

3.	**Coverage and positioning gaps (related work/analysis):**
Several recent LVU/long-video benchmarks and analyses are omitted from Table 1 and the related work (e.g., InfiniBench). Without a broader comparison (domains, clip lengths, task taxonomies, and evaluation protocols), it is hard to judge where VIDEOEVAL-PRO sits in terms of difficulty, realism, and robustness.

--------
**Minor Weaknesses:**

•	Judge circularity/bias: Gemini is used in filtering; GPT-4o-0806 judges answers. Heavy LLM-in-the-loop design risks model-specific biases and version drift.

•	Metric choice: Reporting only Correct rate (omitting F-score) hides behavior on Not-Attempted and may penalize calibrated abstention; short golds (avg 2.1 words) risk lexical-match bias unless synonym canonicalization is detailed.

•	Sampling & budget fairness: “Best over {32…512} frames” mixes sampling policy and capacity; no token/pixel budget normalization or shot/motion-aware sampling ablation is provided.

**Questions:**

1- How stable are results under judge swaps (e.g., Claude, Gemini), and under prompt variations? Can you report accuracy deltas and κ when swapping judges and when removing Gemini from filtering steps?

2- How do you handle synonyms, pluralization, named-entity variants, or minor paraphrases in short answers? Any canonicalization (lemmatization, string rules) or evidence-localized judging to avoid penalizing semantically correct paraphrases?

3- Beyond picking the best frame count per model, can you provide budget-normalized comparisons (e.g., #image tokens or pixels processed), and sampling-policy ablations (uniform vs shot/motion-aware) to decouple model ability from input selection?

4- Since questions originate from prior benchmarks, what audits ensure no textual leakage (e.g., reused phrasings) and that converted items aren’t answerable via dataset priors? Any text-only controls on VIDEOEVAL-PRO and per-source breakdowns?

---

### Note · Authors · 2025-11-12

I have read and agree with the venue's withdrawal policy on behalf of myself and my co-authors.